# A New Mobility Model for Multi-UAVs Reconnaissance Based on Partitioned Zone

**Yong-Il Jo**  **, Muhammad Faris Fathoni and KyongHoon Kim** *

Department of Informatics, Gyeongsang National University, Jinju-daero, Jinju, Gyeongnam 52828, Korea;
crues@gnu.ac.kr (Y.-I.J.); mfarisfathoni@gmail.com (M.F.F.)
* Correspondence: khkim@gnu.ac.kr

**Abstract:** Activities on Unmanned Aerial Vehicle (UAV) have increased over the last years and there are many fields in which UAVs can be used. One of the basic applications is reconnaissance of a given area using multiple UAVs. To perform reconnaissance mission, there are two methods: (i) path planning to navigate the pre-determined route; and (ii) random mobility method to explore without prior knowledge. In this paper, we indicate the imbalance problem of existing random mobility models for reconnaissance and propose a new model considering reconnaissance balance based on the number of visits. We divide the scanning area into $N$ zones and then select a zone stochastically in which the search is insufficient. We evaluated the performance of the model by focusing on the coverage rate and average inter-visiting time. The proposed model shows that the 90%-coverage reaching time is improved by about 25% and the average inter-visiting time is improved by up to 15% compared to the previous approach.

**Keywords:** UAV; mobility model; reconnaissance; pheromone; zone

## 1. Introduction

As computing and communication technologies have developed rapidly, UAVs (Unmanned Aerial Vehicles) have received much attention not only in research fields, but also in real-life applications including forest fire monitoring [1], highway traffic monitoring [2], etc. Since UAVs are operated without human beings, they are useful especially in hazardous or inaccessible environments. Much recent research has focused on operating multiple UAVs in order to accomplish a given task efficiently, such as reconnaissance, target tracking, surveillance, etc. [3–8].

One of important and common missions of multi-UAVs is the cooperative reconnaissance to search a target in a given open area. Thus, many studies have been conducted on efficient cooperative reconnaissance with multiple UAVs [3,4,9–12]. The reconnoitering methods with autonomous flights are classified into the path planning method and the random mobility method [3]. In the path planning method, each UAV routes its own pre-planned path so that UAVs reconnoiter their own determined regions. On the contrary, the random mobility method has no pre-planned paths for UAVs' movements. Each UAV decides its reconnaissance path dynamically based on random mobility models [3,4,9–11,13–16]. This random mobility method can overcome unexpected events, such as UAV failure, and also provides unpredictability of reconnaissance to targets. In this paper, we focus on the random mobility method to enhance the reconnaissance performance.

Although random mobility-based cooperative reconnaissance generally shows good coverage rate, it has one limitation in terms of reconnaissance balance. For example, Random Waypoint [17] shows that most of UAVs' movements are focused on the center area. Pheromone Repel model [9] also shows high reconnaissance rate in a specific area because UAVs fly similarly based on the shared information. Thus, we propose a new random mobility model to overcome the imbalance problem. The proposed

model divides the scanning area into *N* zones and guides UAVs to fly from one border to the other border with a zigzag pattern based on visiting counts.

The remainder of the paper is organized as follows. In Section 2, we deal with related work on mobility models and the research motivation. The proposed model is explained in Section 3. Section 4 presents the simulation environment and the result of the experiment. Finally, Section 5 provides the conclusions and future work.

## 2. Related Work and Research Motivation

### 2.1. Related Work

The path planning-based cooperative reconnaissance decides UAVs' paths in advance in order to reconnoiter a given area [13]. A central planner or controller configures the area and path for each UAV to cover. Let us assume that four UAVs are to reconnoiter a square area. Then, one simple path planning is to command each UAV to cover a quarter of an area evenly. In [18], the authors studied a cooperative path planning with multiple UAVs in an irregular polygonal area. The concave and convex area is partitioned into the same number as UAVs. The UAVs perform searching mission in their own allocated area. When a UAV malfunctiones, the area assigned to each UAV is reconfigured.

In [19–22], the authors studied optimization problems of path planning for the purpose of avoiding some threats. If there is a threatening spot on a planning path by the operator, the UAV avoids it and returns to its original path. On the other hand, if there are no threatening spots, the UAV moves to the path planned by the operator. The algorithms for avoiding threatening spots in path planning method have used Particle Swarm Optimization [20], Ant Colony Optimization [21], Artificial Bee Colony algorithm [22], etc.

This central path-planning method is efficient in terms of coverage rate. However, the method cannot cope with unexpected events such as operational failure of UAVs, dynamic movement of targets, etc. Another weakness of the method is high predictability of reconnaissance path, which enables opponents or targets to avoid the reconnaissance. In addition, as the number of UAVs is increased, it is not a trivial problem to configure the path of each UAV optimally with consideration of many uncertain events.

Another approach of cooperative reconnaissance is the random mobility method in which UAVs reconnoiter without prior knowledge. The random mobility method depends on the mobility model of individual UAV movement. Thus, many recent studies on dynamic reconnaissance of multiple UAVs used mobility models developed in FANET (Flying Ad-hoc Network) or MANET (Mobile Ad-hoc Network) in order to model the UAV movement.

Mobility models can be defined as movement of nodes over time and provide an environment for realistic simulation [23]. In FANET, mobility models are used to generate flight trajectories and evaluate the performance of FANET protocols [24]. To perform the mission of reconnaissance, the Random Waypoint model and the Random Walk models have been used as basic models due to their simplicity [12,25,26]. As shown in Figure 1a, each mobile UAV in Random Waypoint model selects a destination randomly in the area and then moves toward the destination at a speed which is also chosen randomly between 0 and $V_{max}$. In the Random Walk model, each UAV randomly and uniformly chooses its new direction between 0 and 360 degrees, and the new speed between 0 and $V_{max}$. Then, they move for some time interval between 0 and $T_{max}$. While the Random Waypoint model in MANET considers pause time at the destination, the model in FANET does not consider the pause time. Therefore, the two models, i.e. the Random Waypoint model and the Random Walk model, have the same movement in FANET. In this paper, we regard the Random Waypoint model as the same model as the Random Walk model.

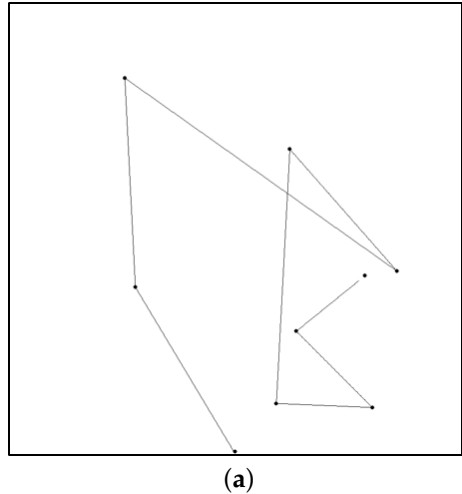 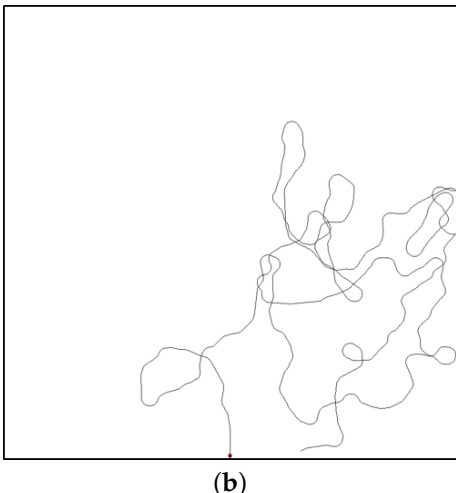

(**a**)        (**b**)

**Figure 1.** The examples of movement: (**a**) Random Waypoint model; and (**b**) Markov Process model.

Figure 1b shows Markov Process model, which is also used as a basic random mobility model for reconnaissance. This model randomly decides one of three directions at every unit time: turning left, flying straight, and turning right. The probabilities of directions are given differently depending on the current status of a UAV. For example, when a UAV flies straight, its next probability of flying straight is 80%, the probability of turning left is 10%, and that of turning right is 10%. Likewise, when a UAV turns left, its next probability of turning left is 90%, the probability of flying straight is 10%, and that of turning right is 0%.

In [9], the random mobility method can be enhanced by sharing reconnaissance information with others. This model uses the pheromones of ants in order to induce a UAV to the area with less pheromone distributions [9]. The pheromone distribution of a UAV is defined by the stored information about paths routed by other UAVs and itself in a bounded region at the current position. UAVs broadcast their own pheromone distributions to other UAVs within a range of communication. When a UAV receives the other UAV's pheromone distributions, it updates its pheromone distributions with the received distribution. Then, each UAV decides its new direction based on the current pheromone distribution in such a way that it avoids the regions which are routed more. If there is no pheromone distribution, a UAV moves based on Markov Process model.

Other models using pheromones include H3MP (Hybrid Markov Mobility Model with Pheromones) [3], Multiple Pheromone UAV Mobility Model [4], etc. H3MP is proposed on a patrol-surveillance scenario for maximizing target detection in a realistic area which is not square. They use the *k*-mean algorithm to divide the area into the similar size. For movement of UAVs, they define the Destination state and the Move state. The Destination state is to choose a destination with transitions probabilities in finite-state machines and the Move state is to search the area based on pheromone. Multiple Pheromone UAV Mobility Model is a reconnaissance model based on the pheromone. It has two kinds of pheromones: repulsive and attractive. Repulsive pheromone is used for searching the area in an ant colony fashion to avoid places which are visited recently. On the other hand, attractive pheromone is used for following a target or to gather surrounding UAVs for tracking a target.

### 2.2. Research Motivation

To analyze the reconnaissance performance, we simulated two well-known mobility models, the Random Waypoint model and the Pheromone Repel model. Random mobility models for reconnaissance can be categorized into two methods according to sharing information. Random Waypoint and Pheromone Repel models are representative models in each category. In our simulations, other random models without sharing information show similar performance as the Random Waypoint.

In the case of models with sharing information, other related work enhancing Pheromone Repel model have different objectives, not just reconnaissance. In terms of reconnaissance, those work perform the same as Pheromone Repel model.

Although previous mobility models show good performance in terms of coverage rate of reconnaissance, they did not consider the balance of the number of visits in each area. In Figure 2, we count and plot the number of visits in each unit area so that the bright color indicates high rate of revisits.

As shown in Figure 2a, the bright-color regions are mostly located at the center of the square, which indicates that most of the reconnaissance are focused in the center of the simulation area. That is, although the UAVs in the Random Waypoint model reconnoiter the area by choosing the destination randomly, they tend to move around the central part of the area.

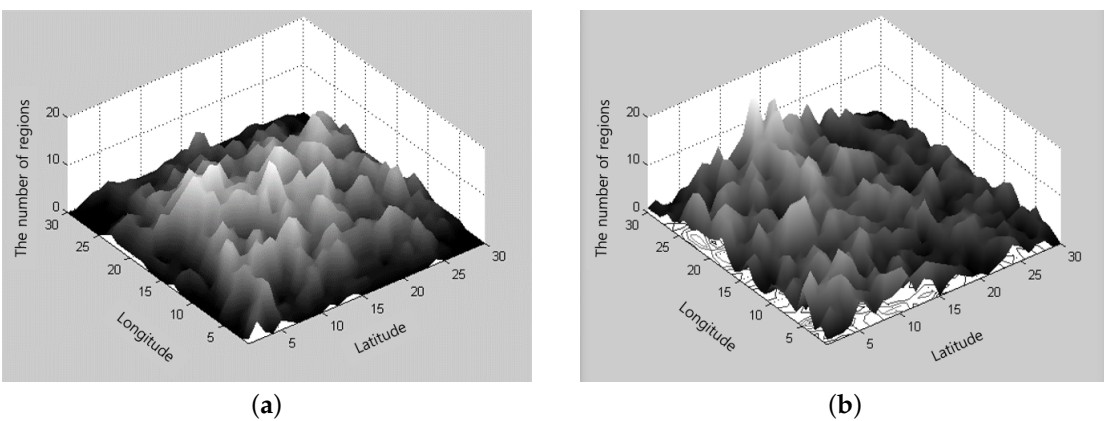

(**a**)　　　　　　　　　　　　　(**b**)

**Figure 2.** The distribution of visits (the bright color indicates the high rate of revisits): (**a**) Random Waypoint model; and (**b**) Pheromone Repel model.

Similarly, the Pheromone model in Figure 2b also shows high revisit rates in some regions. In the case of the Pheromone Repel model, there are two reasons for high revisiting rates in some regions. The first one is due to the Markov model in the absence of any pheromone information. The Markov model leads a UAV to keep turning left (or right) if the UAV turns left (or right). The second reason occurs when two or more UAVs meet. When two UAVs moving in similar direction meet each other, they share their pheromone information. Thus, they move similar ways since they have the same pheromone information.

In Figure 3a, we plot the number of unit regions for each visited counts from the results of Figure 2a. As shown in Figure 3a, the Random Waypoint model reconnoiters 337 unit regions (37.4%) at most twice for 2 h, while it guides UAVs to scan 53 unit regions (5.8%) 10 times or more. Similarly, the Pheromone Repel model also shows imbalance in terms of visiting number among unit regions since some regions are visited even 16 or 17 times, as shown in Figure 3b.

In this paper, we consider this imbalance of visits in each region for designing reconnaissance mobility model. The imbalance of the number of visits affects not only the coverage rate but also the uniform reconnaissance by searching the designated area periodically. Since this imbalance problem becomes worse as the reconnaissance proceeds, we need a new random mobility model to solve it. Therefore, we propose a novel mobility model for the purpose of uniform reconnaissance.

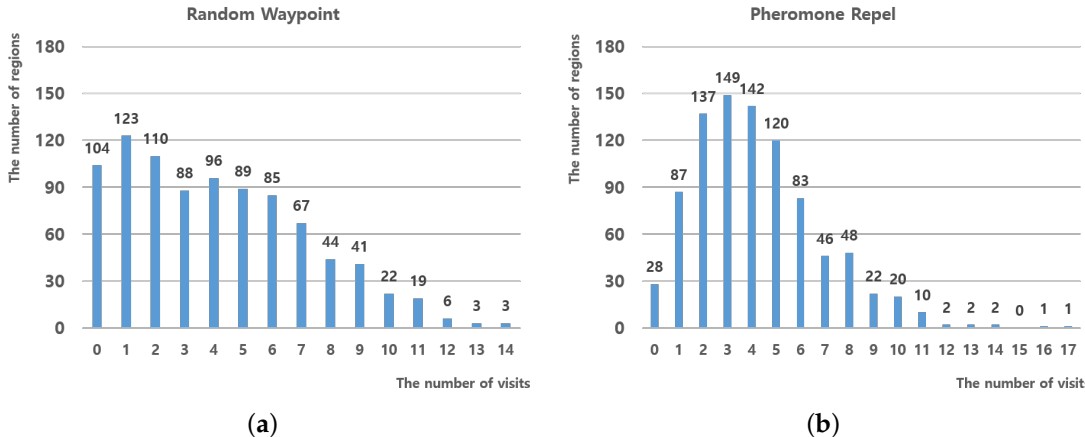

**Figure 3.** The visited counts: (**a**) Random Waypoint model; and (**b**) Pheromone Repel model.

## 3. Proposed Model

### 3.1. UAV Model

In this paper, we assume a fixed-wing aircraft, which is more efficient than rotary-wing aircraft because it searches for a wide area. We assume the simple UAV model used in [9], as shown in Table 1. For realistic motion of fixed-wing UAV's turn, we set rotation radius of 500 m. The UAVs can communicate with others in the range of 8000 m. Let us note that we use *area* for the whole scanning area, *zone* for the unit area of managing waypoints, and *unit region* for the unit area of scanning count throughout this paper.

**Table 1.** Performance of the simple UAV.

| Parameters | Specifications |
| --- | --- |
| Type of aircraft | Fixed-wing |
| Rotation radius | 500 m |
| Communication range | 8000 m |
| Scan area of camera | $2000 \times 1000$ m |
| Flight speed | 150 km/h |

In this paper, we do not consider packet loss in communicating with each other. The UAV is equipped with a camera which can scan a row of 2000 m and a column of 1000 m. The speed of a UAV is fixed at 150 km/h. Lastly, we assume there are no collisions among UAVs by changing altitude.

### 3.2. Random Destination with Pheromone Zone Mobility Model

We enhance the Random Waypoint model by selecting the next destination in the border of the scanning area in order to solve the insufficient exploration at the boundary area in the Random Waypoint model. When UAVs fly from one border to the other border straight, the center area is still frequently reconnoitered. Thus, the proposed model guides the UAV to move in a zigzag pattern toward the destination in the border area. This zigzag pattern makes the target difficult to predict the route and results in the visiting rate in each area to be possibly even, as well. We manage the visited count in a zone with a suitable size and induce UAVs to follow the zones with lower visited count to the destination in the border.

The brief algorithm of the proposed mobility model is shown in Algorithm 1. A UAV selects a border zone and flies to the zone through several waypoints. When the UAV arrives at the intermediate waypoints, it selects a next waypoint in the next zone. In addition, the UAV updates its pheromone

information about intermediate waypoints and shares the information with other UAVs. Thus, the model consists of three procedures: (1) selecting border zone; (2) selecting next zone and waypoint; and (3) exchanging information. In the procedure of selecting border zone, we describe the types of zone such as the border zone and the immediate zone, and the method of selecting border zone. The procedure of selecting next zone includes the method of establishing the intermediate zone in order to have the zigzag pattern until reaching the border zone. In the procedure of exchanging information, we explain how to share the pheromone information between UAVs. Based on the shared information, the UAV stochastically selects the next zone in which reconnaissance is insufficient.

---

**Algorithm 1:** UAV Mobility Model

---

/* - $P_{curr}$: the current position of the UAV
　- $Z_{curr}$: the current zone of the UAV
　- $W_z$: the set of waypoints visited in the zone $z$
　- $Ph_{info}$: the pheromone information or the list of $W_z$
*/
Initialize $P_{curr}$ and $Z_{curr}$ as a starting position.
**while** *true* **do**
　$Z_{dest} \leftarrow selectNextBorderZone(Z_{curr})$
　**do**
　　$Z_{next} \leftarrow selectNextZone(Z_{curr}, Z_{dest}, Ph_{info})$
　　$W_{next} \leftarrow selectWaypoint(Z_{next})$
　　**while** $P_{curr} \neq W_{next}$ **do**
　　　Control the UAV to move to $W_{next}$
　　**end**
　　$W_z \leftarrow W_z \cup \{W_{next}\}$
　　$Z_{curr} \leftarrow Z_{next}$
　**while** $Z_{curr} \neq Z_{dest}$;
**end**

---

### 3.2.1. Selecting Border Zone

In the proposed method, the area to be scouted is divided into $N$ zones. Each UAV stochastically selects and moves to the next zone which has been visited less than other zones nearby. As we assume the whole area to be a rectangular shape, the area is divided into $N(=M_1 \times M_2)$ zones. An operator configures the size of a zone according to the rotation radius of a UAV. If the distance to the destination in the next zone from the current position is shorter than the rotation diameter, the UAV may not reach the destination by keeping circuitous flight around the point. To prevent the possible circuitous flight, we decide the minimum size of a zone to be greater than the rotation diameter. For example, when the size of the simulation area is 10 km $\times$ 10 km and the maximum rotation diameter of the UAV is 1 km, the maximum number of zones which can be divided is 100 (=10 $\times$ 10).

The zones are classified into *border zones* which are located in the boundary area of the scanning region, and *intermediate zones* for other zones inside. For example, the scanning area in Figure 4 is divided into 16 zones from $z_1$ to $z_{15}$. In Figure 4a, intermediate zones include $z_5$, $z_6$, $z_9$, and $z_{10}$, while the other zones belong to border zones.

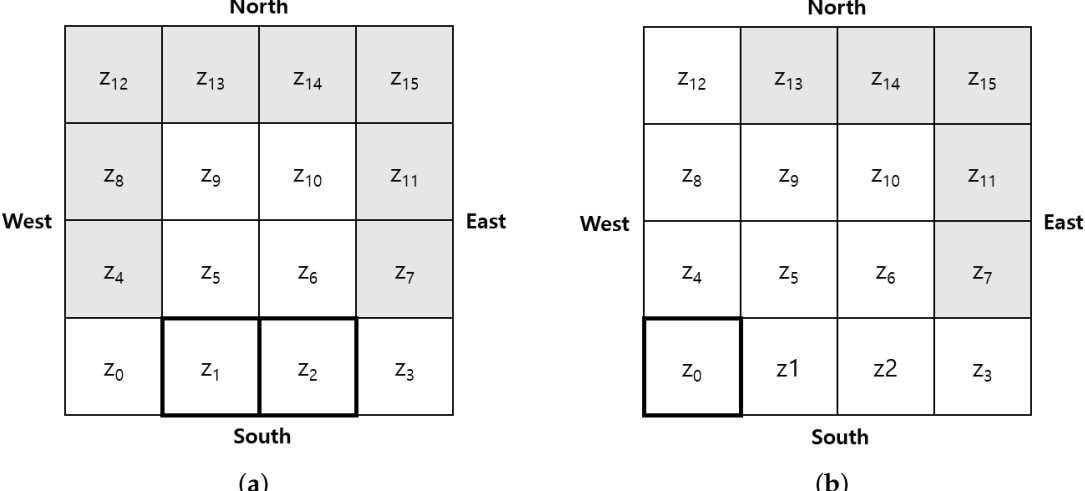

**Figure 4.** The examples of selecting candidates for destination border zone (shading squares are candidates for destination border zone): (**a**) the candidate border zones of $z_1$ and $z_2$; and (**b**) the candidate border zones of $z_0$.

A UAV selects the next destination border zone in the other sides from the current boundary. If the current zone is located in $z_1$ or $z_2$, as shown in Figure 4a, the possible destination border zones are the eight shaded zones. On the other hand, if the current zone is located in the edge of the scanning area such as $z_0$, the candidates of destination border zone are $z_7$, $z_{11}$, $z_{13}$, $z_{14}$, and $z_{15}$, as shown in Figure 4b. As $z_0$ belongs to both the western border zone and the southern border zone, those border zones are excluded when a UAV decides a destination border zone.

When the destination border zone is decided, we consider the number visited in each zone so that a higher probability is assigned to the zone with lower number of visits. Let us denote $N_z$ as the number of visited waypoints in a zone. For a given set of border zones, $B$, the probability of selecting a border zone $z$ is defined by:

$$P_z = \begin{cases} \dfrac{N_{total} - N_z}{(|B| - 1) \times N_{total}} & \text{if } N_{total} \neq 0 \\ \dfrac{1}{|B|} & \text{if } N_{total} = 0 \end{cases} \tag{1}$$

where $N_{total}$ is the total number of waypoints in all zones in $B$.

Let us consider Figure 5 as an example, where the current zone is $z_2$ and the six dots indicate the waypoints previously visited. The set of candidate zones, B, becomes $\{z_4, z_7, z_8, z_{11}, z_{12}, z_{13}, z_{14}, z_{15}\}$. The total number of waypoints visited in $B$ is 6. Table 2 shows the probabilities of eight candidate border zones according to Equation (1). The probability of selecting $z_4$ is the lowest because it has the highest number of visited waypoints. On the contrary, zones with no visited waypoints, such as $z_7$, $z_{12}$, and $z_{13}$, have higher probabilities than others.

**Table 2.** The probabilities of selecting border zones in Figure 5.

| | The Probability of Selecting Zone | | | | | | | |
|---|---|---|---|---|---|---|---|---|
| **Zone** | $z_4$ | $z_7$ | $z_8$ | $z_{11}$ | $z_{12}$ | $z_{13}$ | $z_{14}$ | $z_{15}$ |
| **Probability** | $\dfrac{3}{42}$ | $\dfrac{6}{42}$ | $\dfrac{5}{42}$ | $\dfrac{5}{42}$ | $\dfrac{6}{42}$ | $\dfrac{6}{42}$ | $\dfrac{5}{42}$ | $\dfrac{6}{42}$ |

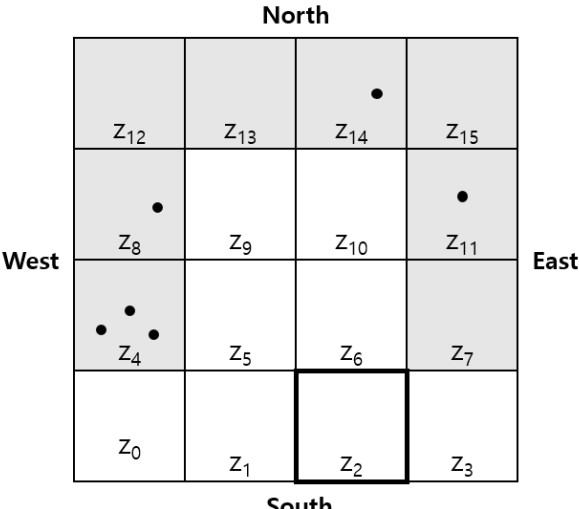

**Figure 5.** An example of waypoints in the border zone of $z_2$.

### 3.2.2. Selecting Next Zone and Waypoint

As explained in Algorithm 1, a UAV does not fly to the destination border zone directly, but tries to pass by zones with lower visited waypoints. Thus, we need a mechanism to select the next zone. First, we decide the candidate set of next zones from the current zone. A neighbor zone is called a candidate zone if its Manhattan distance or L1 distance to the destination border zone is shorter than that of the current zone. For example, let us assume that a UAV with the destination border zone $z_{15}$ is now in $z_1$, as shown in Figure 6a. Then, the candidate next zones are $z_2$, $z_5$, and $z_6$. Similarly, Figure 6b,c shows the candidate next zones at each zone while the UAV moves.

Next, we decide the next zone from the candidate next zones based on the number of waypoints visited previously. Similar to the equation of deciding the destination border zone, the probability of selecting the next zone is inversely proportional to the number of waypoints visited in the zone. Thus, for a given set of candidate next zones $C$, the probability of selecting the next zone $z$ is given by Equation (2).

$$P_z = \begin{cases} 1 & \text{if } |C| = 1 \\ \dfrac{N_{total} - N_z}{(|C| - 1) \times N_{total}} & \text{if } |C| > 1 \text{ and } N_{total} \neq 0 \\ \dfrac{1}{|C|} & \text{if } |C| > 1 \text{ and } N_{total} = 0 \end{cases} \qquad (2)$$

In Figure 6a, let us assume that the numbers of waypoints in the three candidate next zones ($z_2$, $z_5$, and $z_6$) are known as 5, 3, and 1, respectively. Since the $N_{total}$ (= 9) is not zero, we calculate the probability of selecting the next zone by using Equation (2). As shown in Table 3, each probability of moving to $z_5$ is 6/18, $z_6$ is 8/18, and $z_2$ is 5/18. That is, the $z_6$ is more likely to be selected because it has fewer visits than the other two zones. Figure 6d presents a newly added waypoint in each zone and a moving path by a UAV. As a UAV is passed through zones $z_6$, $z_{10}$, and $z_{15}$, each zone has a new waypoint by the UAV. Thus, $z_6$ has two waypoints and the other zones, $z_{10}$ and $z_{15}$, have three waypoints.

Table 3 shows the probabilities of selecting the next zone in each step assuming that the UAV selects $z_6$ and $z_{10}$. If the UAV chooses $z_{14}$ or $z_{11}$ in Step 3 as the next zone, the zone to be chosen in the next step will be the destination zone because $|C|$ is 1.

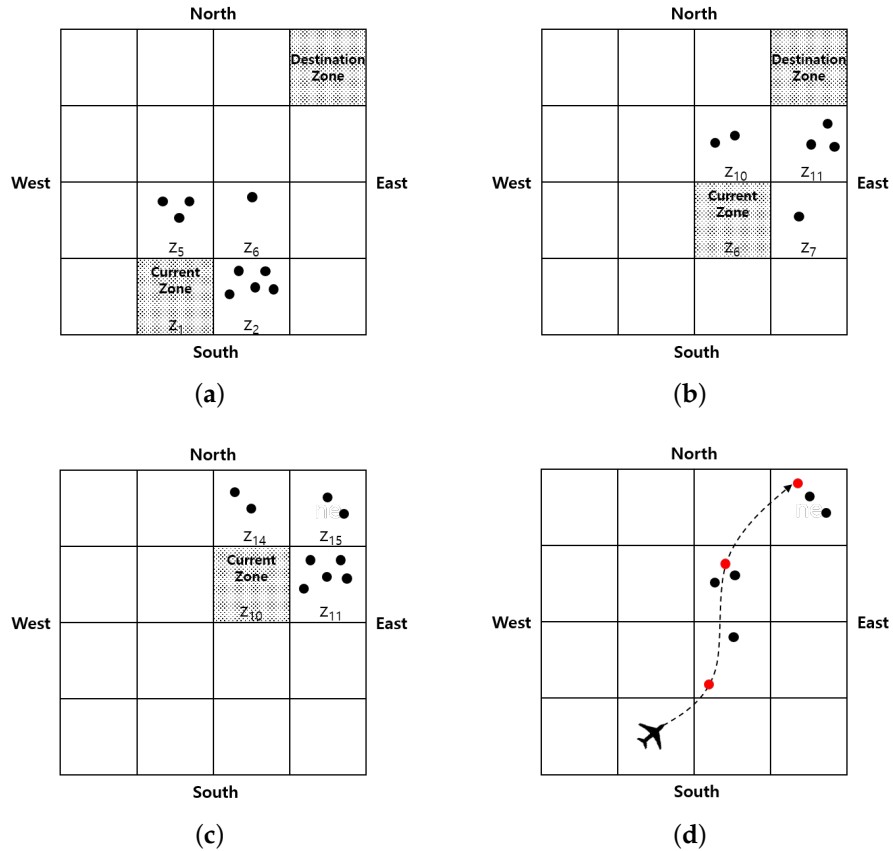

**Figure 6.** The examples of selecting next zone: (**a**) Step 1 ($C = \{z_1, z_5, z_6\}$); (**b**) Step 2 ($C = \{z_7, z_{10}, z_{11}\}$); (**c**) Step 3 ($C = \{z_{11}, z_{14}, z_{15}\}$); and (**d**) a path moved to the destination zone.

**Table 3.** The probabilities of selecting next zones in Figure 6.

| | | | | | | | | | |
|---|---|---|---|---|---|---|---|---|---|
| **The Probability Table** | | | | | | | | | |
| | | **Step 1** | | | **Step 2** | | | **Step 3** | |
| **Zone** | | $z_2$ | $z_5$ | $z_6$ | $z_7$ | $z_{10}$ | $z_{11}$ | $z_{11}$ | $z_{14}$ | $z_{15}$ |
| **Probability** | | $\dfrac{5}{18}$ | $\dfrac{6}{18}$ | $\dfrac{8}{18}$ | $\dfrac{5}{12}$ | $\dfrac{4}{12}$ | $\dfrac{3}{12}$ | $\dfrac{4}{18}$ | $\dfrac{7}{18}$ | $\dfrac{7}{18}$ |

Once the next zone is determined, the UAV decides the waypoint in the next zone so that it flies to the waypoint. One constraint for the waypoint is the minimum distance from the current position is more than the rotation radius in order to prevent a circuitous flight.

### 3.2.3. Exchanging Information

We define the zone waypoint information of a zone $z$ as the set of waypoints selected by UAVs in the zone $z$, and denote it as $W_z$. When a UAV arrives at the waypoint $p$ in the next zone $z$, it updates $W_z$ by adding the point $p$. The zone waypoint information should also v updated to other UAVs. Thus, each UAV broadcasts the zone waypoint information periodically, which is called as the pheromone information in [9]. In this work, we define the pheromone information as the list of all zone waypoint information managed by a UAV.

Let us denote the zone waypoint information of $z$ managed by a UAV $u$ as $W_z^u$. When a UAV $u$ receives $W_z^v$ from a UAV $v$, $W_z^u$ is updated as $W_z^u \cup W_z^v$. That is, the UAV adds the new waypoints in the zone $z$, which are not in its own waypoint information.

## 4. Performance Evaluation

We evaluated performance comparison of the Random Waypoint model, the Pheromone Repels model, and the proposed model for the UAV model in Table 1. Let us note that we used the parameters in Table 1 since other simulation results show similar tendency for other UAV parameters. Table 4 shows other parameters used in simulations. Ten UAVs reconnoitered in the area of 30 km × 30 km for 2 or 3 h. We divided the scanning area into 15×15 zones. We implemented a reconnaissance model simulator using C++ language, and measured the average performance by repeating 20 times in each case.

**Table 4.** Simulation parameters.

| Parameters | Specifications |
| --- | --- |
| UAVs | 10 ea |
| Simulation area size | 30,000 × 30,000 m |
| Simulation time | 7200, 10,800 s |
| zones | 15 × 15 |

We measured the coverage rate and the averaged inter-visiting time. The coverage rate was defined by the rate of scanning the area in the whole area. The inter-visiting time was the time interval between two consecutive visits in a unit region. Thus, the average inter-visiting time of the set of visited unit regions, $V_R$, for the simulation time $T_{sim}$ is defined by Equation (3).

$$T_{avg} = \frac{1}{|V_R|} \sum_{r \in V_R} \frac{T_{sim}}{N_r} \tag{3}$$

where $N_r$ is the number of visits in unit region $r$. The lower inter-visiting time indicates that UAVs visit the unit region frequently. Thus, we used the average inter-visiting time as a measurement of how evenly UAVs reconnoiter the area.

### 4.1. Coverage Comparison

Figure 7 shows the experimental results simulated for 7200 s and 10,800 s, which are the times when the proposed model and the Random Waypoint reached the steady states, respectively. The proposed model divides the area into 15 × 15 zones, which are appropriate size of the proposed model. We explain this in Section 4.3.

As shown in Figure 7a, the proposed model provides better results than the other models in terms of the coverage rate. The initial coverage rate is similar, but the times to reach 80% and 90% rate are different, as shown in Table 5. The Random Waypoint model could not satisfy more than 90%, even though simulation time is 7200 s. The 90% reaching time of the Pheromone Repel model is 4193 s while the proposed model takes 3173 s. That is, the proposed model is about 1021 s faster than the Pheromone Repel model for covering 90%, which means 24% performance improvement. Similarly, the 80%-coverage reaching time of the proposed model is reduced by 41.7% and 17.9% compared to the Random Waypoint and the Pheromone Repel models, respectively.

In Figure 7b, the Random Waypoint model attains 90% coverage rate at 6472 s, which takes more than two times as long as the proposed model. We can see that the Random Waypoint model is not only slow to reconnoiter the new unit region but also scanned the same area repeatedly. The 90%-coverage reaching time of the Pheromone Repel model takes 4208 s and the proposed model takes 3122 s, which means the proposed model has a performance improvement of about 25% compared to the Pheromone Repel model. The main reason of this performance improvement is because the proposed model moves between border zones trying to pass intermediate zones with lower visited numbers.

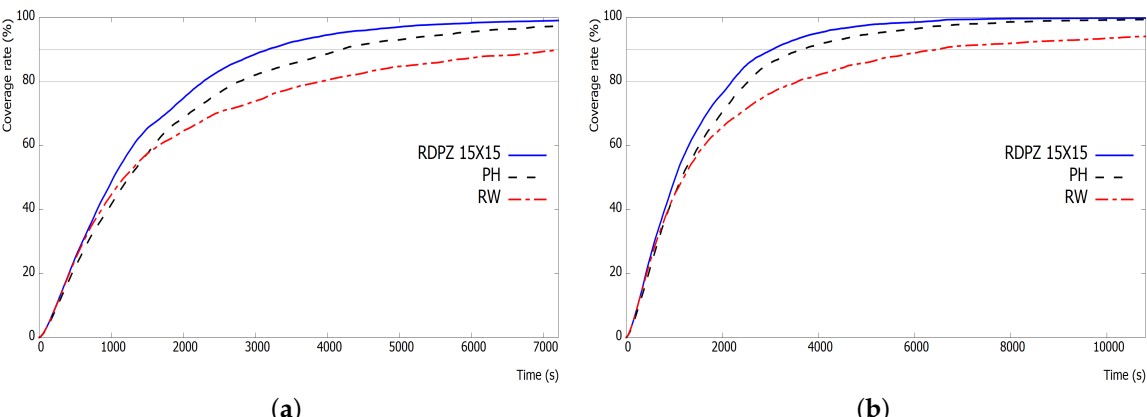

**Figure 7.** The coverage rate results: (**a**) 7200 s; and (**b**) 10,800 s.

**Table 5.** The 80%- and 90%-coverage reaching time of each model.

|  | The Mobility Model | The 80% Reached Time (s) | The 90% Reached Time (s) |
|---|---|---|---|
| **7200 s** | Random Waypoint | 3908 | - |
|  | Pheromone Repel | 2775 | 4194 |
|  | Random Zone 15 × 15 | 2276 | 3173 |
| **10,800 s** | Random Waypoint | 3872 | 6472 |
|  | Pheromone Repel | 2817 | 4208 |
|  | Random Zone 15 × 15 | 2249 | 3122 |

### 4.2. Average Inter-Visiting Time Comparison

We evaluated the average inter-visiting time of reconnoitered area in order to check whether UAVs scout the area uniformly or not. Figure 8 shows the average inter-visiting times of three models for 2 and 3 h. As shown in Figure 8a, the proposed model shows 18.5% and 9.4% lower inter-visiting time than the Random Waypoint model and the Pheromone Repel models, respectively. The proposed model shows 34% and 15.1% less than the two other models in 3-h simulations. In addition, the proposed model in 3-h simulation shows less inter-visiting time than that of 2-h simulation, which implies the proposed model makes UAVs scan the area evenly in the steady state. On the contrary, the Random Waypoint model has higher inter-visiting time in the case of the 3-h simulation because UAVs fly mostly in the center area.

We also plot the visited numbers of the unit region (1 km × 1 km) in Figure 9a in order to analyze the uniformity of coverage. Let us note that the coverage of the Random Waypoint model is distributed in center of simulation area in Figure 2a, and the Pheromone Repel model has high coverage on a particular region in Figure 2b. However, the proposed model shows uniform search compared to the other two models.

Figure 9b shows the distribution of regions according to the number of visits. The maximum number of visits in unit region is 10, while the two other models show 14 and 1, as shown in Figure 3. To analyze the skewness of visits, we plot the number of regions per the number visits with the connected lines in Figure 10. We also draw the mean, median, and mode (most frequent value) of the three distributions. As shown in Figure 10, the proposed model is close to the Gaussian distribution so that mean, median, and mode are almost same in the center. On the contrary, the two other models show left-skewed distributions, which indicates much imbalance among regions in terms of number of visits.

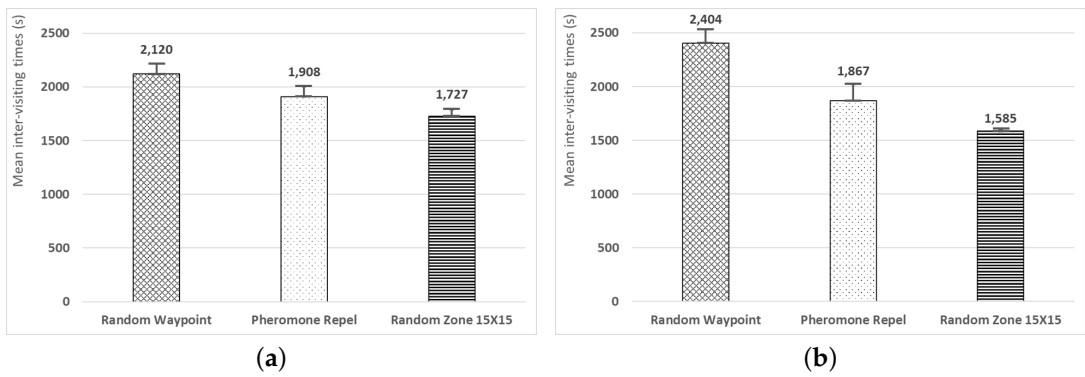

**Figure 8.** The average inter-visiting times: (**a**) 7200 s; and (**b**) 10,800 s.

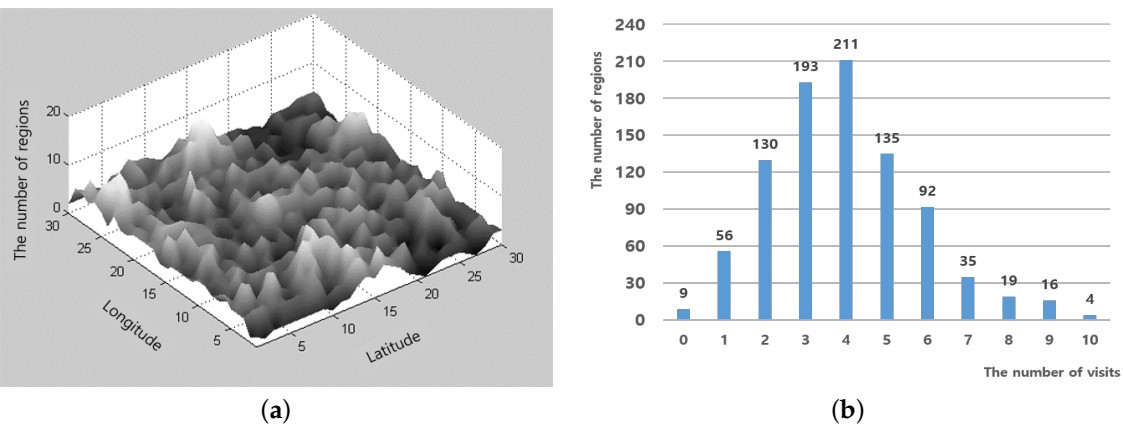

**Figure 9.** The results of RDPZ: (**a**) the coverage distribution; and (**b**) the number of the visits in each unit region.

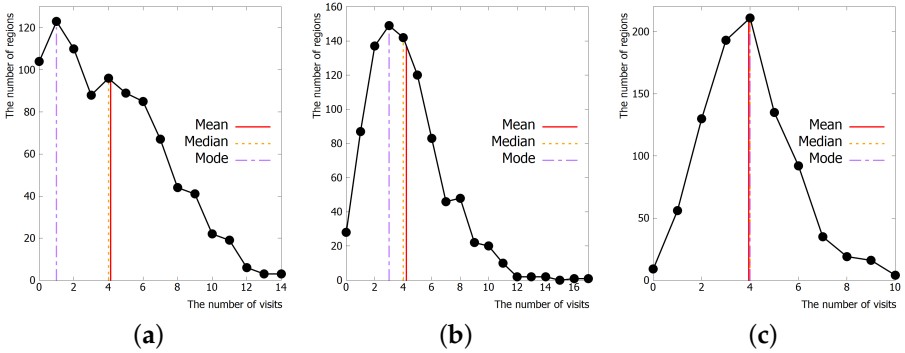

**Figure 10.** The comparisons of skewness: (**a**) Random Waypoint model; (**b**) Pheromone Repel model; and (**c**) RDPZ 15 × 15 model.

### 4.3. Coverage Rate Comparison According to the Size of Simulation Area

In this subsection, we evaluate the effect of the zone size according to the scanning area. We simulated the proposed model for 20 km × 20 km, 30 km × 30 km, and 40 km × 40 km areas, respectively. Figure 11 shows the coverage rate results of three different area. In each area, we also varied the number of zones from 5 × 5 to 40 × 40. As shown in Figure 11, the number of zones in each area affects the coverage rate. The best numbers of zones in 20 km × 20 km, 30 km × 30 km, and 40 km × 40 km areas are 10 × 10, 15 × 15, and 20 × 20, respectively. That is, the appropriate size of a single zone is 2 km × 2 km assuming that the parameters of UAVs are given in Table 1.

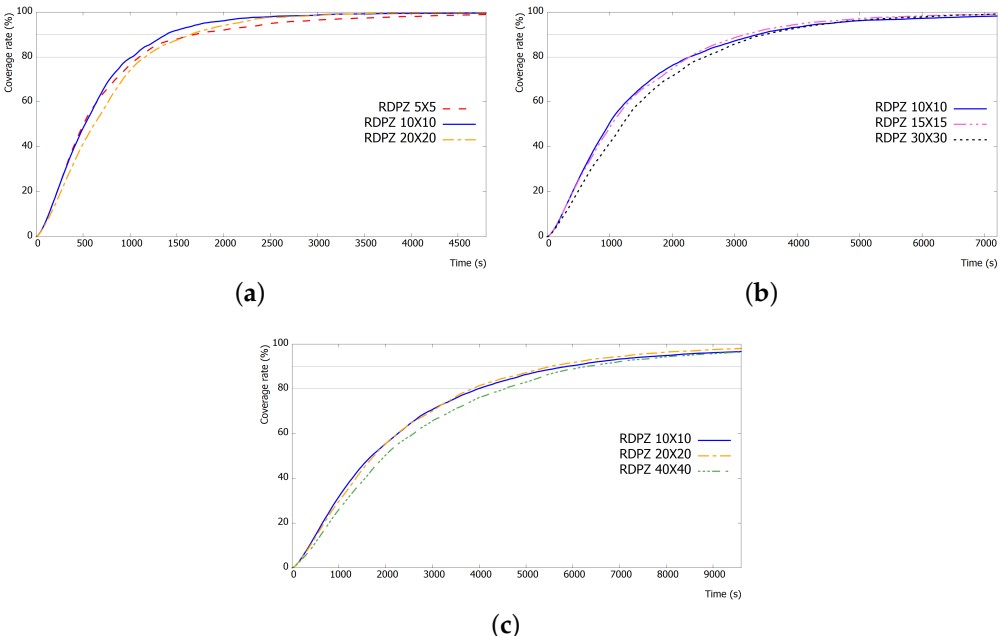

**Figure 11.** The coverage rate results: (**a**) 20 km × 20 km area; (**b**) 30 km × 30 km area; and (**c**) 40 km × 40 km area.

## 5. Conclusions

In this work, we proposed the new mobility model, RDPZ (Random Destination with Pheromone Zone), for reconnaissance with multi-UAVs. The RDPZ mobility model guided uniform reconnaissance by dividing the scanning area into $N$ zones in order to solve the imbalance problem. To evaluate the proposed model, we measured the coverage rate and inter-visiting time of the Random Waypoint model, the Pheromone Repel model, and RDPZ model. When we simulated for 10,800 s, 90%-coverage reaching time was about 51.7% better than the Random Waypoint model and about 25% better than the Pheromone Repel model. Similarly, the average inter-visiting time was about 34% better than the Random Waypoint model and about 15.1% better than the Pheromone Repel model. In addition, the experimental results according to the size of the scanning area show the RDPZ models with the half size of the scanning area has the best performance.

Our on-going work includes enhancing the proposed model in irregular shaped areas including hazard region. In addition, we are also planning to analyze the performance results (e.g., the effect of reconnaissance time) in order to enhance the proposed model. We will apply target tracking, collision avoidance, etc. to RDPZ model and compare other mobility models. After that, we will redesign the mobility model by focusing on different environments.

**Author Contributions:** Y.-I.J. and K.H.K. proposed the mobility model. Y.-I.J. implemented the model, conducted the experiments, and wrote manuscript under the supervision of K.H.K., M.F.F assisted and performed model comparison experiments. All the authors have reviewed and revised final manuscript.

**Acknowledgments:** This work was supported partly by the Gyeongsang National University Fund for Professors on Sabbatical Leave, 2017, and by the Human Resources Development of the Korea Institute of Energy Technology Evaluation and Planning (KETEP) grant funded by the Ministry of Trade, Industry and Energy (No. 20194030202430).

**Conflicts of Interest:** The authors declare no conflict of interest.

## Abbreviations

The following abbreviations are used in this manuscript:

| Symbol | Description |
|---|---|
| $P_{curr}$ | the current position of the UAV |
| $Z_{curr}$ | the current zone of the UAV |
| $W_z$ | the set of waypoints visited in the zone $z$ |
| $Ph_{info}$ | the pheromone information or the list of $W_z$ |
| $Z_{dest}$ | the destination zone |
| $Z_{next}$ | the selected next zone |
| $W_{next}$ | the selected next waypoint |
| $P_z$ | the probability of selecting a border zone or a next zone |
| $B$ | the set of border zone |
| $N_z$ | the number of visited waypoints in a zone |
| $N_{total}$ | the number of visited waypoint in all zones |
| $C$ | the set of candidate next zones |
| $W_z$ | the number of waypoints in a zone |
| $W_z^u$ | the number of waypoints in UAV $u$ |
| $W_z^v$ | the number of waypoints in UAV $v$ |
| $T_{avg}$ | the average inter-visiting time |
| $V_r$ | the set of visited unit regions |
| $T_{sim}$ | the simulation time |
| $N_r$ | the number of visits in unit region $r$ |
| $z_1, z_2, \cdots, z_{15}$ | the example names of each zone |

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
