# Peer review of "A New Mobility Model for Multi-UAVs Reconnaissance Based on Partitioned Zone"

_applsci, doi:10.3390/app9183810_

Round 1

Reviewer 1 Report

The manuscript titled ‘A New Mobility Model for Multi-UAVs Reconnaissance based on Partitioned Zone’ presents a simulation study that tests the performance of a new movement model for multi-unmanned aerial vehicle (UAV) missions to optimize reconnaissance coverage. The algorithm underlying the movement model is described well and readers will be able to translate it into their own work. The work is conducted to a satisfactory standard. I would like to see the following points addressed before the work can be published though.

General comments:

Why did the authors choose the Random Waypoint and Pheromone Repulsion models for comparison? Are these recognized as the best? In line 111 you state that they are well known, but the reader does not know whether these are basic models that perform badly compared to other existing models, or whether they are recognized as having high performance for coverage rate and inter-visiting time. This information is critical for a reader to decide how important the current findings are likely to be.

Details of how the simulations were carried out (e.g., software, programming language, etc.) are completely absent. It also seems like each simulation was repeated only once, giving no indication as to how reliable the estimates of performance are likely to be.

What happens if you vary the parameters of the UAV from Table 1? Is this result likely to be generalizable to other practitioners, or are will this movement model perform poorly in some circumstances?

In lines 51-56 you introduce avoidance of areas. Your algorithm will be affected if such a condition is introduced, but this is not discussed at all.

Specific comments:

L111-113 The information in this sentence needs to be in the figure caption for Figure 2. If looking at Figure 2 without reading the main text, the reader has no idea what the different shading means.

L117-119 Do you have data to support this claim besides your subjective interpretation of how this pattern arose? Could some areas have high revisit rates because in the absence of any pheromone information, the Markov model led them to do a high proportion of left turns (given there is a 90% chance that they will continue to turn left if they are currently turning left) or alternatively right turns? Under this scenario, they would stay in a similar area for longer, thereby revisiting some locations more.

Figure 5 More detail is needed in the caption. The reader has to decide through intuition what the shading represents.

Figure 6 The caption needs to indicate that this is an example dataset.

L195-197 change ‘A neighbor zone is called the candidate next zone if the Manhattan distance or L1 distance to the destination border zone is shorter than the current zone’ to ‘A neighbor zone is called a candidate zone if its Manhattan distance or L1 distance to the destination border zone is shorter than that of the current zone’. The current wording makes the reader think that there is only one candidate when there can be multiple.

L218-219 Constraining where the next waypoint can be by placing it not less that the rotation radius is necessary, but in some cases, it may require the UAV to stray outside of the boundaries of the scanning area. For example, if in Figure 7c Z14 was chosen instead of Z15, the point would have to be on the northern side of Z14 to satisfy the turn radius condition. Thus, when flying north into Z14, the UAV would not be able to turn rapidly enough to remain inside the boundaries of the scanning area in order to then turn into Z15. This possibility needs to be discussed.

L244 10,800 needs to be followed by ‘seconds’.

L258 Change ‘25% rather than Pheromone Repel model’ to ‘25% compared to the Pheromone Repel model’

Figure 9 The y-axis needs an axis label (e.g., Mean inter-visiting time (s)) and error bars depicting either standard error, standard deviation, or similar should be added to each column.

L269 Change ‘the Random Waypoint shows more in case of 3-hour simulation because UAVs’ to ‘the Random Waypoint model has higher inter-visiting time in the case of the 3-hour simulation because UAVs’.

L274-275 Change ‘However, the proposed model shows uniform search rather than two models’ to ‘However, the proposed model shows uniform search compared to the other two models’.

Figure 11 An x-axis labels is required (e.g., Number of visits).

Figure 12 Can the legend items be listed in order of increasing or decreasing cell size rather than at random as they appear at present? Also, could the colours and type of line (solid/dashed) be kept consistent for each zone size (i.e. all 20 km X 20 km lines kept solid blue, and all 10 km X 10 km lines be kept dashed black).

L288 You state ‘As shown in Figure 12, the zone size affects the coverage rate.’ However, looking at Figure 12, I would argue that zone size has little effect on coverage rate. The effect is gone by the time the simulations reach ~7000 seconds. This could be discussed more thoroughly in relation to how someone using your method could determine what would be the optimum zone size for their purposes. For example, it looks like small zone sizes (relative to the size of the scanning area) are best if reconnaissance times are short, whereas large zone sizes are possibly better if reconnaissance times are longer.

L289 Is ‘The best zone size is generally 2km×2km’ supposed to be ‘The best zone size is generally 20 km×20 km’? Also, 20 km x 20 km is plotted in only two of the panels on this figure, and arguably it is best in only one of these two cases. I think this statement needs to be reconsidered or discussed in more detail. In your conclusion, you state that zone sizes that are about half the size of the scanning area produce optimal results. That is a much better statement than the arbitrary one you provided in L289.

Reviewer 2 Report

Generally the paper is well written. Some observations and comments:

lines 17-18: "Much recent research has focused..." yet the papers refereed to are from 2016 and earlier? lines 109-111: "Although previous mobility models show good performance in terms of coverage rate of reconnaissance, they did not consider the balance of the number of visits in each area.We simulated two well-known mobility models" a) Selection of models to investigate and compare would perhaps benefit from clearer comparison and overview b) why exactly two? Why those two? Fig. 2: names and units on axes are better than just an uninformative letter (x,y,z). Same also Fig. 10. Fig. 4: not all used notations are described, not even in Table A1. Same also elsewhere (zones M1 and M2 etc.) line 244: Why 7200 and 10800? Why 15x15? Fig. 8: would be nice to have mentioned 80% and 90% lines present. Same also on Fig 12. Fig. 9: Units on vertical axes are missing. Fig. 11: Horizontal axis undefined.

Presented results are based on simulation. It is perhaps tricky to conduct real experiments, but it would be quite interesting to see them, and compare results with the simulation.
